# CAT: Coordinating Anatomical-Textual Prompts for Multi-Organ and Tumor Segmentation

**Zhongzhen Huang**[1,2], **Yankai Jiang**[2], **Rongzhao Zhang**[2],
**Shaoting Zhang**[1,2✉], **Xiaofan Zhang**[1,2✉]

[1]Qing Yuan Research Institute, Shanghai Jiao Tong University   [2]Shanghai AI Laboratory

{huangzhongzhen,xiaofan.zhang}@sjtu.edu.cn,
{jiangyankai, zhangrongzhao, zhangshaoting}@pjlab.org.cn

## Abstract

Existing promptable segmentation methods in the medical imaging field primarily consider either textual or visual prompts to segment relevant objects, yet they often fall short when addressing anomalies in medical images, like tumors, which may vary greatly in shape, size, and appearance. Recognizing the complexity of medical scenarios and the limitations of textual or visual prompts, we propose a novel dual-prompt schema that leverages the complementary strengths of visual and textual prompts for segmenting various organs and tumors. Specifically, we introduce *CAT*, an innovative model that **C**oordinates **A**natomical prompts derived from 3D cropped images with **T**extual prompts enriched by medical domain knowledge. The model architecture adopts a general query-based design, where prompt queries facilitate segmentation queries for mask prediction. To synergize two types of prompts within a unified framework, we implement a ShareRefiner, which refines both segmentation and prompt queries while disentangling the two types of prompts. Trained on a consortium of 10 public CT datasets, *CAT* demonstrates superior performance in multiple segmentation tasks. Further validation on a specialized in-house dataset reveals the remarkable capacity of segmenting tumors across multiple cancer stages. This approach confirms that coordinating multimodal prompts is a promising avenue for addressing complex scenarios in the medical domain. Codes are available at https://github.com/zongzi3zz/CAT.

## 1  Introduction

Advanced prompt engineering [1, 2] has augmented large language models [3, 4, 5] with emerging capabilities. However, these paradigms have not yet been successfully applied to vision tasks, primarily due to the ever-changing and unpredictable natures of computer vision [6, 7]. This is particularly evident in medical domains, where variations in imaging protocols, noise/artifacts, and patient-specific pathologies pose significant challenges [8, 9, 10]. To tackle these challenges, there have been some efforts within the community to develop promptable models for segmenting objects in medical images: 1) One type of such effort focuses on textual-prompted models [11, 12, 13], which show profound competencies in segmenting a specific organ or tumor referenced by arbitrary text phrases. These approaches involve distilling knowledge from language models like CLIP [14] or BERT [15] to facilitate the alignment between visual and textual representations; 2) Another direction in promptable segmentation research aims to visual-prompted models [16, 17, 18, 19, 20, 21], which rely on visual examples or visual landmarks (e.g., boxes and points). Recently, the trend toward visual-prompted segmentation models in the medical domain mainly focuses on fine-tuning SAM [22] with lightweight, plug-and-play adapters or modifying SAM into 3D-based architecture.

---

[✉]Corresponding authors.

38th Conference on Neural Information Processing Systems (NeurIPS 2024).

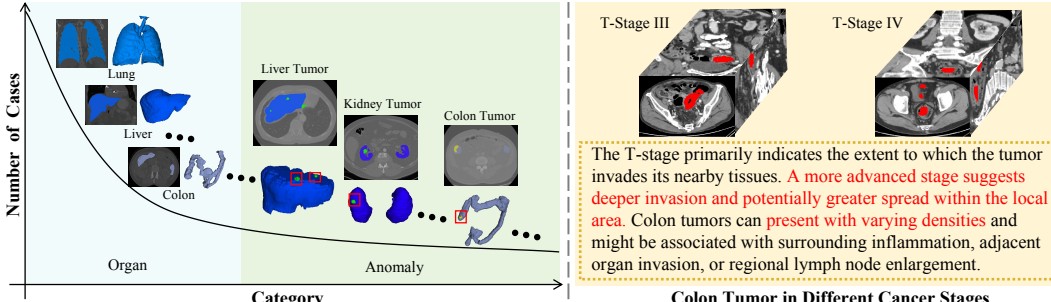

Figure 1: Left: Long-tailed curve of the category and the number of available cases that can be obtained in the medical field. Right: Tumors in different cancer staging with diverse shapes and sizes.

Despite the promising advancements brought by the prompt-enhanced segmentation paradigm, those segmentation models still face significant limitations. As illustrated in the left part of Figure 1, the distribution of medical datasets often follows a long-tail pattern, where an increase in the diversity of detectable anomalies corresponds with a sharp decline in the number of available cases. Textual-prompted methods utilize textual representations from referred text phrases to guide the segmentation process, requiring alignment between visual and textual representations. Although descriptive texts can cover intricate and rare anomalies with domain knowledge, data scarcity due to long-tailed distribution hinders the effective learning of alignments between textual and visual representations. This issue is particularly significant in the medical domain, where numerous corner cases (e.g., tumors with variations in shape, size, density distribution, and blurring boundaries) need to be addressed. Conversely, visual prompts are not constrained to the need for cross-modal alignment, providing a more intuitive and direct method to enhance the segmentation process. Nonetheless, visual prompts fail to convey the general concept of each object, leading to a performance drop when confronted with various scenarios in medical domains, especially for tumors. For instance, colon tumors range from different cancer stages [23] and have significant inter- and intra-patient in tumor sizes and shapes, as shown in the right part of Figure 1. Moreover, tumors at the same stage can exhibit varying densities. Given the inherent diversity in tumors, there is a necessity to provide comprehensive knowledge of each tumor type via textual descriptions.

In this work, we strive to develop a promptable segmentation model that utilizes the strengths of both visual and textual prompts without human interaction, aiming at a fully automatic model for medical professionals. We propose a new dual-perspective prompting scheme. On the one hand, we directly employ the cropped volumes derived from the anatomical structure as our visual prompts. We refer to such prompts as anatomical prompts, intending to represent target objects in a more intuitive and visually coherent manner. On the other hand, we enhance the textual prompts with more comprehensive knowledge. With the proposed prompting scheme, we introduce *CAT*, a model towards comprehensive segmentation that harnesses a dual-prompting mechanism to **C**oordinate **A**atomical and **T**extual prompts.

*CAT* follows the general query-based design with two extra parallel encoders dedicated to processing anatomical and textual prompts into prompt queries. Inspired by previous segmentation models [24, 25, 26, 27], we adopt query embeddings to perform mask predictions. A ShareRefiner is utilized to refine segmentation queries and prompt queries by attending them to image features from the backbone. We employ two distinct feature assignment strategies for prompt queries, ensuring that these prompt queries are disentangled for more versatile representations. Following the refinement process, PromptRefer updates segmentation queries by integrating both types of prompt queries. The mask prediction module then generates binary masks by performing a simple dot-product of segmentation query embeddings with the pixel embedding map obtained from the backbone. To further facilitate the coordination, we augment *CAT* with cross-modal alignment between anatomical and textual prompts. By training on an assembly of 10 public CT datasets in the abdomen, *CAT* demonstrates strong segmentation capabilities on these datasets and achieves remarkable results on an in-house dataset encompassing four cancer stages. Further ablation analysis shows that anatomical and textual prompts serve complementary roles. To summarize, our contributions are threefold:

- We design a new prompting scheme that utilizes both complementary strengths of anatomical prompts and textual prompts for medical image segmentation.
- We build *CAT*, a fully automatic and promptable model that coordinates anatomical prompts derived from 3D cropped volumes with textual prompts enriched by medical domain knowledge, enabling strong flexibility for various segmentation tasks.
- Extensive experiments demonstrate the benefits of coordinating anatomical prompts and textual within one model. *CAT* consistently achieves state-of-the-art performance on multiple segmentation tasks and has generalization capability to diverse tumor types.

## 2 Related Work

### 2.1 Textual-prompted Segmentation in Medical Imaging

Prompt engineering has [1, 2] achieved remarkable progress in natural language processing and shown great potential in general visual perception [28, 29, 30, 24, 7, 31]. By leveraging pre-trained vision-language foundation models [14], textual-prompted methods demonstrate impressive capabilities in open-vocabulary segmentation [31, 32, 33, 34, 35, 36]. These advancements in the natural image field have significantly propelled the field of 2D medical image segmentation [37], where the innovative model architecture is used to recognize a variety of visual contexts through textual prompts. Recently, some efforts tried to train a textual-prompted universal model for segmenting various organs and tumors in 3D volumes [13, 12, 11]. A pre-trained text encoder is adopted to encode the injected prompts and guide the target grounding process. Moreover, the latest attempts [12] focused on extending the short textual phrases with knowledge. However, the reliance on texts struggles with medical image segmentation due to the potential misalignment between textual descriptions and complex visual patterns. To solve the limitations caused by linguistic ambiguity, we integrate visual inputs (i.e., anatomical prompts) for more accurate and comprehensive image perception.

### 2.2 Visual-prompted Segmentation in Medical Imaging

Beyond textual-prompted methods, the field has seen a notable shift towards incorporating visual prompts to enhance accuracy and context sensitivity [16]. Different from textual prompts, visual prompts introduce more flexibility and context-awareness for models. The innovative Segment Anything [22] introduces a promptable model for generalized image segmentation. Building upon the foundation model, the field has seen a substantial shift towards applying this paradigm to the medical domain. One line adapts SAM to general medical image segmentation with fine-tuning [17, 19, 20, 21, 18]. Some works like SAM-Med2D [19] and 3DSAM-adapter [20] utilize adapters [38] to transfer the capabilities of SAM to medical images with a few number of trainable parameters. Other line [39] trains 3D models from scratch straightforwardly. SAM-Med3D [40] and SegVol [39] developed 3D SAM models from scratch by reforming the 2D model and training with a large scale of CT scans. Incorporated with a higher annotated ratio, a 3D point-promptable model CT-SAM3D [41] is introduced with a progressively and spatially aligned prompt encoding technique. Nonetheless, due to the substantial disparities among anomalies in the medical domain, visual prompts could not provide a generic concept for each type. Our work resembles the textual and anatomical prompts to support the multi-organ and tumor segmentation.

## 3 Method

In this paper, we focus on applying anatomical prompts (i.e., cropped 3D medical volumes) and textual prompts for generic segmentation tasks in the abdomen involving organs and tumors. *CAT* employs a popular query-based encoder-decoder architecture with a sophisticated interaction paradigm between queries and prompts, aiming at predicting $N$ categories in the abdomen, as shown in Figure 2. Our model integrates four main components: i) Vision Backbone, designed to extract image features and construct pixel embedding maps, ii) Prompt Encoders, used to encode anatomical and textual prompts provided by users, respectively, iii) ShareRefiner, utilized to refine segmentation queries and prompt queries, and iv) PromptRefer, generating target queries for prediction.

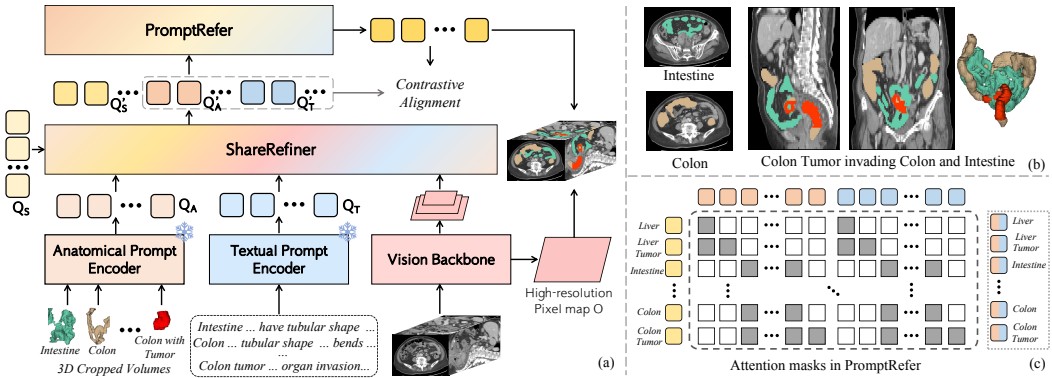

Figure 2: (a) **CAT** follows the query-based segmentation architecture. 3D cropped volumes according to the anatomical structure are utilized as anatomical prompts. Texts enhanced by professional knowledge are adopted as textual prompts. Learnable queries and both prompts are utilized for the final prediction via ShareRefiner and PromptRefer. (b) The case of colon tumor in Stage-IV invading the intestine. (c) Attention masks in PromptRefer for assigning specific prompts to queries.

## 3.1 Vision Backbone and Prompt Encoders

**Backbone.** Following other unified models [26, 42] for segmentation tasks, we perform mask classification for all organs and tumors. We adopt a key idea from Mask2Former [42, 12] to construct a pixel embedding map, which is obtained from the backbone encoder features. As shown in Figure 2, the vision backbone comprises a vision encoder **Enc** responsible for extracting multi-scale visual features $\mathbf{V}$ and a vision decoder represented as **Dec**, which gradually upsamples visual features to the high-resolution pixel embedding map $O$. Given an input 3D volume $I \in \mathbb{R}^{H \times W \times D}$, the process can be formally represented as follows:

$$\mathbf{V} = \mathbf{Enc}(I), O = \mathbf{Dec}(\mathbf{V}), \tag{1}$$

where $\mathbf{V} = \{V^i\}_{i=1}^{L}$, $V^i \in \mathbb{R}^{H^i \times W^i \times D^i \times C^i}$, and $L$ is the number of layers of the vision encoder. Here, $H^i, W^i, D^i$ and $C^i$ denote the height, width, depth and channel dimension of $V^i$, respectively. $O \in \mathbb{R}^{H \times W \times D \times C_o}$ is obtained by fusing feature maps from the **Enc**, $C_o$ is the dimension of $O$.

**Prompt Encoders.** Prompts have been widely used in segmentation tasks recently. Textual prompts provide comprehensive concepts for the target object but bring some ambiguities, such as the fact that both the intestines and the colon have a "tubular shape". Conversely, visual prompts are more helpful in disambiguating the user's intent when textual prompts fail to identify the correct object. However, these visual prompts have limited robustness in handling large variations in medical images [41]. Our method enhances the segmentation process via the coordination of anatomical and textual prompts. Specifically, the anatomical prompt is a referred region from another CT exam, which can be a particular tumor or a cropped volume of the same semantic concept. Given $N$ cropped volumes $P_{Aj} \in \mathbb{R}^{H_A \times W_A \times D_A}, j \in \{1, 2, ..., N\}$, we encode these volumes through a pre-trained encoder $\mathbf{Enc}_A$ and obtain anatomical prompt embeddings $\mathbf{E}_A$. To convey the comprehensive notion of targets with detailed information, we utilize long textual descriptions as our textual prompts. Following previous work [12], we incorporate textual prompts with medical domain knowledge. We employ the text encoder $\mathbf{Enc}_T$ [43] to encode these descriptions and use the [CLS] token output as the textual prompt embedding, denoted as $\mathbf{E}_T$. The process can be denoted as follows:

$$\mathbf{E}_A = \mathbf{Enc}_A(\mathbf{P}_A), \mathbf{E}_T = \mathbf{Enc}_T(\mathbf{P}_T) \tag{2}$$

$$\mathbf{Q}_A = \mathrm{Linear}(\mathbf{E}_A), \mathbf{Q}_T = \mathrm{Linear}(\mathbf{E}_T), \tag{3}$$

where $\mathbf{Q}_A, \mathbf{Q}_T \in \mathbb{R}^{N \times C}$ are prompt queries and $C$ is the channel dimension.

## 3.2 ShareRefiner

**CAT** predicts $N$ masks based on the learnable segmentation queries $\mathbf{Q}_S$ and the pixel embedding map $O$. Our initial steps involve feeding both segmentation queries $\mathbf{Q}_S$ and prompts queries $\mathbf{Q}_A, \mathbf{Q}_T$

into the proposed ShareRefiner to enable the queries to interact with multi-scale features. In our approach, queries will selectively attend to local image features for self-refinements. Specifically, the ShareRefiner consists of a series of cross-attention blocks where queries perform cross-attention with respect to the target visual feature $V^i$. To disentangle these queries, we adopt soft and hard assignments for $[\mathbf{Q}_S, \mathbf{Q}_T]$ and $\mathbf{Q}_A$, respectively. A hard cross-attention layer is applied to extract anatomical prompt features from the multi-scale feature maps, ensuring that each anatomical query gathers discriminative visual regions without overlaps. For the $i$-th layer of ShareRefiner, the hard assignment similarity matrix of $\mathbf{Q}_A^i$ is computed via Gumbel-Softmax [44, 45] as follows:

$$S^i = \mathbf{Q}_A^i \, \mathrm{Flatten}(V^i)^T \tag{4}$$

$$S_{\mathrm{gumbel}}^i = \mathrm{Softmax}\left(\left(S^i + G^i\right)/\tau\right), \tag{5}$$

$$S_{\mathrm{onehot}}^i = \mathrm{Onehot}\left(\mathrm{argmax}_N\left(S_{\mathrm{gumbel}}^i\right)\right). \tag{6}$$

Where $S^i, G^i \in \mathbb{R}^{N \times H^i W^i D^i}$, $G^i$ are i.i.d random samples drawn from the $Gumbel(0, 1)$ distribution and $\tau$ is a learnable coefficient. The one-hot operation of the $\mathrm{argmax}$ is performed over $S_{\mathrm{gumbel}}^i$ for hard assigning. Since the straightforward hard assignment (i.e., one-hot) is not differentiable, the straight-through trick in [46, 47] is used to compute the assignment similarities $S'^i$ of one-hot value:

$$S'^i = \left(S_{\mathrm{onehot}}^i\right)^\top + S_{\mathrm{gumbel}}^i - \mathrm{sg}\left(S_{\mathrm{gumbel}}^i\right), \tag{7}$$

where sg denotes the stop gradient operator. Following the feature grouping process, we use a self-attention layer to regulate the relationships among queries and a feed-forward layer for projection. Formally, the aforementioned process is represented as:

$$\mathbf{Q}'_S, \mathbf{Q}'_T = \mathbf{ShareRefiner}([\mathbf{Q}_S, \mathbf{Q}_T], \mathbf{V}, \mathrm{hard} = \mathrm{False}), \tag{8}$$

$$\mathbf{Q}'_A = \mathbf{ShareRefiner}(\mathbf{Q}_A, \mathbf{V}, \mathrm{hard} = \mathrm{True}). \tag{9}$$

### 3.3 PromptRefer

In practice, when segmenting target objects, more attention needs to be paid to the relevant context. Localizing the typical tumor requires being aware of the anomalous features in the relevant organ, and even identifying organs requires focusing on the anatomical structures involved. For instance, as shown in Figure 2(b), colon tumors of Stage-IV would invade the adjacent organs like intestines. Directly combining prompt queries for mask prediction is suboptimal since there is still an inherent gap between the two types of prompt embeddings even though we have unified them into prompt queries. The typical objective is to classify $\mathbf{Q}'_S$ into respective regions with the guidance of prompts. To this end, we introduce **PromptRefer** implemented by the cross-attention mechanism with carefully crafted attention masks in Figure 2(c). Here, a group of prompt queries $\{Q'_{Ai}, \ldots, Q'_{Tj}, \ldots\}$ is employed to a specific segmentation query. To ensure the distinction of queries, segmentation queries are required to only interact with queries in the group. This is illustrated in the following equation:

$$\mathbf{O}_S = \mathbf{PromptRefer}([\mathbf{Q}'_S, \mathbf{Q}'_A, \mathbf{Q}'_T], \mathrm{mask}). \tag{10}$$

Here, $\mathbf{O}_S$ represents the decoded segmentation query features for predicting masks.

To further integrate two types of prompts and push segmentation queries closely to the referenced prompt, we employ query-level contrastive learning in our model. Specifically, given the decoded segmentation query features and refined prompt queries, we define two types of losses and calculate the InfoNCE loss [14] as follows:

$$\mathcal{L}_{s2p} = -\frac{1}{N}\sum_{i=1}^{N} \frac{\exp(\tilde{O}_{Si} \cdot \tilde{Q}'_{\{A/T\}i})}{\sum_{j=0}^{N}\exp(\tilde{O}_{Si} \cdot \tilde{Q}'_{\{A/T\}j})}, \mathcal{L}_{p2p} = -\frac{1}{N}\sum_{i=1}^{N} \frac{\exp(\tilde{Q}'_{Ai} \cdot \tilde{Q}'_{Ti})}{\sum_{j=0}^{N}\exp(\tilde{Q}'_{Ai} \cdot \tilde{Q}'_{Tj})}. \tag{11}$$

Where $\tilde{\mathbf{O}}_S, \tilde{\mathbf{Q}}'_A, \tilde{\mathbf{Q}}'_T$ are derived via linear projection layers. The contrastive alignment can be regarded as a distillation process, whereby each prompt query contributes to the segmentation queries. It also benefits the knowledge exchange between queries from two modalities.

### 3.4 Training Objective and Strategy

**Training Objective.** The binary mask proposals $\mathbf{M} \in [0, 1]^{N \times H \times W \times D}$ are calculated through the multiplication operation between the decoded segmentation query features $\mathbf{O}_S$ and the high-resolution pixel embedding map $O$ followed by a Sigmoid. We employ the dice loss for mask

prediction. For classification loss, following [12, 13], we adopt the cross-entropy loss that measures the difference between predicted objects and the categories. Moreover, the contrastive loss is applied to the similarities among queries. The final loss takes the following form:

$$\mathcal{L}_{total} = \mathcal{L}_{dice} + \mathcal{L}_{cls} + \mathcal{L}_{s2p} + \mathcal{L}_{p2p}. \tag{12}$$

**Anatomical prompt training strategy.** For anatomical prompts, we leverage the bounding box derived from masks in the public dataset and relevant anatomical structures to crop a set of prompt volumes for each category and unify these volumes into the same size. To alleviate the domain shift, we use a pre-trained encoder [48] for feature extraction. During training, we randomly sample an instance from the set of prompt volumes for each category, excluding the identical one.

**Textual prompt training strategy.** For the textual prompt, we identify the category names in the abdomen. We call GPT-4 [49] to generate descriptions with medical domain knowledge for each category more than 20 times. We recruit one board-certified physician to rewrite the description according to the results. Besides long descriptions for each category, we also construct several short textual templates. We use the long description for the positive category and randomly sample short phrases for those negative categories. More details can be seen in the Appendix A.1.

## 4 Experiments

**Dataset and Settings.** *CAT* is trained on the curated dataset from 10 public datasets [50, 51, 52, 53, 54, 55, 56, 57, 58, 59], which contain multiple organs and tumors in the abdomen. Following settings in [13], we assemble this dataset with the data pre-processing to reduce the domain gap among various datasets. In the testing phase, two public datasets and one in-house dataset are used for evaluation. The in-house test dataset contains 80 3D CT volumes with colon tumor masks, ranging from Stage-I to Stage-IV. For evaluating the accuracy of organ segmentation, we employ FLARE22 [60] as the external test set. For tumor segmentation, we utilize the combination of MSD dataset [59] and the private dataset. Details of datasets are in Appendix A.2.

**Implementation Details and Evaluation Metrics.** Our model follows the query-based segmentation architecture [42, 26]. That is, we use Swin UNETR [61] as the backbone and Clinical-Bert [43] as the text-encoder. We follow the experiment settings of previous work [13, 12]. Details are in Appendix A.3. For the evaluation, the Dice Similarity Coefficient (DSC) and 95% Hausdorff Distance (HD95) are utilized to gauge the performance of organ/tumor segmentation.

### 4.1 Main Results

**Organ Segmentation.** In this study, we explore the organ segmentation performance of our model on the external testing set of FLARE22. We present the detailed organ-wise segmentation results in Table 1. Compared to models [17, 19] adapting SAM [22] to the medical fields, our approach yields substantially better performance across all 12 organs. For instance, *CAT* significantly surpasses MedSAM [17] by a large margin of 21% DSC points on the adrenal gland segmentation and 15% DSC points on the esophagus segmentation. Such results show the gap in applying 2D SAM-based models for 3D volumes, especially for objects with small sizes or intricate shapes. When compared to 3D SAM models [41, 39, 40], our model performs significantly better than models [39, 40], which take in visual prompts like SAM. For example, we surpass SAM-Med3D [40] and SegVol [39] by 30% points and 20% points in terms of the average score. We can observe that these models also struggle with the problem of handling some difficult organs. In addition, just with one prompt for each category, *CAT* achieves comparable results with CT-SAM3D [41], which utilizes spatially aligned prompts progressively in multi-rounds. Owing to the PromptRefer module, our model demonstrates superior performance in organs where tumors frequently occur. Specifically, *CAT* leads by 2% points in the liver and by 6% points in the pancreas. *CAT* also demonstrates superior performance compared with recent textual-prompted models [13, 12]. Although the gap of organ segmentation is only 2% points, we observed that they do not generalize well on the tumor segmentation task without more intuitive visual prompts, as described in the following part. More details are shown in Table 6.

**Tumor Segmentation.** We do the evaluation on MSD dataset [59] and an in-house dataset to validate its tumor segmentation capability. Table 2 shows the comparison results of four tumor categories. We do not report HD95 scores for SAM-based methods since this comparison may not be entirely

Table 1: Organ segmentation performance on FLARE22. The results(%) are evaluated by DSC. Scores of SAM-based are adopted from the CT-SAM-Med3D [41]. † denotes obtained via the official pre-trained weights. ∗ means implemented from the official code and trained on the same dataset. Abbreviations: "Liv."-Liver, "R_Kid."-Right Kidney, "Spl."-Spleen, "Pan."-Pancreas, "Aor."-Aorta, "IVC"-Inferior Vena Cava, "RAG"-Right Adrenal Gland, "LAG"-Left Adrenal Gland, "Gal."-Gallbladder, "Eso."-Esophagus, "Sto."-Stomach, "Duo."-Duodenum, " L_Kid."-Inferior Vena Cava.

| Methods | Liv. | R_Kid. | Spl. | Pan. | Aor. | IVC | RAG | LAG | Gal. | Eso. | Sto. | Duo. | L_Kid. | Avg. |
|---|---|---|---|---|---|---|---|---|---|---|---|---|---|---|
| SAM [22] | 86.0 | 87.6 | 84.5 | 53.4 | 77.5 | 44.5 | 19.4 | 33.9 | 52.4 | 35.2 | 68.0 | 44.4 | 82.6 | 59.2 |
| MedSAM [17] | 93.0 | 90.0 | 89.1 | 73.5 | 82.5 | 76.5 | 36.0 | 48.7 | 56.4 | 64.7 | 84.0 | 53.9 | 89.7 | 72.2 |
| SAM-Med2D [19] | 91.4 | 83.7 | 83.9 | 58.8 | 60.6 | 18.6 | 10.6 | 27.1 | 32.9 | 28.1 | 72.9 | 45.4 | 86.0 | 53.8 |
| SAM-Med3D [40] | 85.4 | 84.2 | 84.7 | 46.9 | 60.4 | 44.5 | 32.6 | 35.3 | 56.0 | 32.6 | 46.9 | 27.4 | 84.9 | 55.5 |
| SegVol [39] | 83.9 | 71.7 | 75.9 | 69.4 | 83.1 | 80.3 | 42.1 | 49.7 | 55.6 | 69.6 | 81.1 | 55.6 | 75.1 | 68.7 |
| CT-SAM3D [41] | 95.6 | 95.0 | 96.1 | 83.6 | **94.5** | **91.8** | **78.4** | **82.5** | **88.4** | **82.9** | **92.3** | 73.2 | 94.8 | **88.4** |
| Universal† [13] | 97.4 | 95.5 | 96.4 | 73.7 | 84.9 | 84.4 | 72.9 | 73.4 | 86.0 | 76.8 | 88.5 | **74.5** | 96.9 | 84.7 |
| ZePT∗ [12] | 96.7 | 95.6 | 96.6 | 84.3 | 90.0 | 84.4 | 67.2 | 66.8 | 79.6 | 74.2 | 85.2 | 59.1 | 97.2 | 82.8 |
| CAT | **97.7** | **96.3** | **97.1** | **89.2** | 90.5 | 88.0 | 73.6 | 74.3 | 83.0 | 80.1 | 88.2 | 73.4 | **97.3** | 86.8 |

Table 2: Segmentation performance (%) of tumors on MSD [59] and In-house dataset. We compare our method with traditional and promptable methods. † denotes obtained via the official pre-trained weights. ∗ means implemented from the official code and trained on the same dataset.

| Method | MSD Dataset (Tumor in Abdomen) | | | | | | | | In-house Data (Colon Tumor) | | | | | |
|---|---|---|---|---|---|---|---|---|---|---|---|---|---|---|
| | Liver | | Pancreas | | Hepatic Vessel | | Colon | | T1 | T2 | T3 | T4 | Avg. | |
| | DSC↑ | HD95↓ | DSC↑ | HD95↓ | DSC↑ | HD95↓ | DSC↑ | HD95↓ | DSC↑ | | | | DSC↑ | HD95↓ |
| nnUNet∗ [62] | 66.42 | 42.29 | 43.50 | 25.80 | 66.90 | 47.59 | 41.41 | 153.06 | 19.51 | 45.06 | 44.87 | 45.54 | 43.00 | 150.48 |
| Swin UNETR∗ [48] | 68.67 | 42.54 | 41.77 | 22.87 | 63.32 | 44.02 | 39.35 | 161.26 | 21.40 | 33.32 | 46.11 | 52.72 | 45.92 | 168.25 |
| SAM-Med3D† [40] | 44.78 | - | 40.05 | - | 44.86 | - | 39.23 | - | 34.28 | 42.65 | 50.20 | 42.65 | 47.11 | - |
| SegVol† [39] | 66.20 | - | 46.36 | - | 68.57 | - | **60.63** | - | **36.93** | 42.63 | **60.17** | 49.83 | 50.28 | - |
| Universal† [13] | 65.68 | 63.31 | 45.72 | 16.58 | 66.31 | 51.47 | 42.26 | 115.40 | 7.11 | 43.28 | 46.52 | 53.08 | 47.14 | 140.28 |
| ZePT∗ [12] | 68.58 | 43.23 | 44.39 | 19.47 | 68.12 | 33.94 | 40.38 | 113.07 | 23.87 | 34.64 | 50.81 | 51.09 | 46.28 | 155.83 |
| CAT | **72.73** | **34.64** | **49.67** | **15.56** | **70.11** | **33.44** | 48.31 | **108.26** | 30.62 | **45.61** | 55.85 | **57.37** | **53.35** | **80.96** |

equitable. Our ***CAT*** outperforms baselines [62, 48] significantly across all tumor subtypes, achieving at least a 4% improvement in DSC. For the comparison of SAM-based methods [40, 39], it is worth noting that only one point derived from the ground truths is utilized as the prompt during the inference time since tumors of small sizes are difficult to identify but can be easily segmented with the provided prompts. Without using information from labels, ***CAT*** still achieves better performance except for the colon tumor segmentation, as evidenced by a 5% improvement in the DSC of the pancreas tumor. These results validate the proposed PromptRefer module (i.e., attending to specific organs when segmenting tumors). Moreover, we compare ***CAT*** with textual-prompted methods. ***CAT*** surpasses the previously across four tasks by 4% in the average DSC. As our model is an extension of textual-prompted methods, these results demonstrate that incorporating the anatomical prompts can produce higher-quality masks for tumors. More details are shown in Table 7.

We further conduct the evaluation on an in-house colon tumor dataset, with tumors ranging from Stage-I (T1) to Stage-IV (T4) according to the Cancer Staging System [23]. As shown in Table 2, ***CAT*** showcases strong capabilities in dealing with tumors of diverse sizes and shapes. To provide deeper insight, we present the average score of each subtype. It is noted that the higher the number after the "T", the larger the tumor or the more extensively it has grown into nearby tissues. We can observe that SAM-derived methods perform well in segmenting relatively small T1 tumors with point prompts. However, SegVol [39], which achieves the best results in MSD colon tumor segmentation, struggles to handle tumors that invade nearby organs or tissues. It is evident from the performance gap between T3 and T4 tumors. In these cases, tumors in each instance can present with varying densities and show distinct shapes at different locations. Only using visual prompts without comprehensive descriptions to cover all abnormal parts proves challenging. Such scenarios are vital in the medical field, highlighting the need for further research in applying visual prompts for tumor segmentation. In this paper, we coordinate anatomical and textual prompts along with the predefined attention mask to alleviate this problem. ***CAT*** outperforms other models by at least absolute 7% DSC in T4 and 3% DSC on average, demonstrating much better generalizability and robustness. This result suggests that leveraging anatomical prompts together with medical domain knowledge is an alternative way to address intricate scenarios in the medical domain.

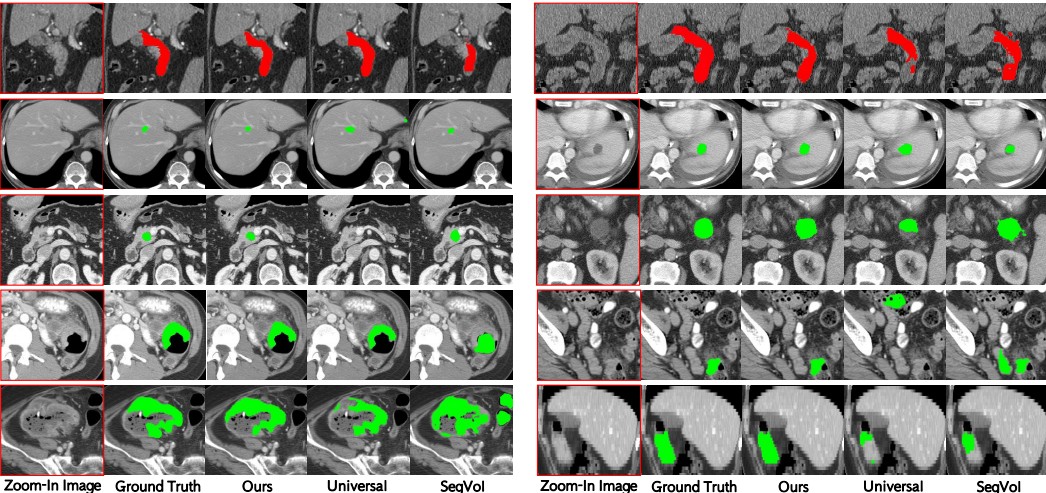

Figure 3: Qualitative visualizations of the proposed model and other prompting methods on or-gan/tumor segmentation. The segmentation results presented from rows one to five correspond, in order, to the duodenum, liver tumors, pancreas tumors, colon tumors, and colon tumors in Stage-IV.

**Qualitative Comparison.** To give more intuitive comparisons, we present the qualitative results in Figure 3. In the first row, we observe that current methods may struggle with segmenting tubular-shaped organs, particularly SAM-derived methods like SegVol [39], since visual prompts can not convey the general concept of such tubular-shaped objects. The results in the second and third rows reveal that textual-prompted models are prone to misclassification and overlook key details due to the lack of intuitive context, underscoring the importance of visual prompts. However, these only relying on visual prompts struggle with tumors of intricate shapes without the support of medical domain knowledge, as shown in the fourth row, where SegVol exhibits a high number of false positives on normal CT scans. When tumors invade other tissues, existing models often fail due to segmentation target incompleteness and misclassification of normal regions as tumors. In contrast, ***CAT*** can precisely identify most abnormal regions and consistently generate results that are more consistent with the ground truth compared to all other models.

## 4.2 Ablation Studies

To fully investigate the contribution of our introduced components, we conducted ablation studies to compare each part in both organ and tumor segmentation tasks. We selected a subset of organ and tumor categories that are particularly challenging for models, facilitating a more intuitive comparison.

**Ablation of Two Prompts.** We start from a basic query-based segmentation architecture without any extra prompts, and then we ablate the effectiveness of using different prompts for different tasks. As demonstrated in Table 3 (the first four rows), eliminating the prompt schema leads to a substantial performance drop on both organ and tumor segmentation ($64.08 \rightarrow 54.39$ in Duodenum and $72.49 \rightarrow 66.37$ in Liver tumor). The observed declines in the third row indicate that textual features, which are rich in semantics but lack appearance features, may not generalize well to objects with intricate structures or small sizes. It is evidenced by a performance reduction of $6\%$ in Esophagus segmentation and $4\%$ in Hepatic Vessel tumor segmentation. Moreover, we can observe from the second row that the tumor segmentation capability is inadequate when relying solely on anatomical prompts. This issue stems from the diversity and variance of tumors. For instance, every example the model encounters is drastically different as it tries to identify the location and nature of pancreas tumor. Although tumors can be regarded as anomalies, the lack of consistent context poses a challenge for the model in establishing a general concept solely with the cropped volumes. The coordination of anatomical and textual prompts improves the performance of the promptable model. This confirms the efficacy of our joint prompts schema, which is designed to help the model form more stable and generalizable masks in the medical field.

Table 3: Ablation studies of two prompts and model designs on organ and tumor segmentation dataset.

| Variant | | | | Organ (%) | | | | | Tumor (%) | | | | |
|---|---|---|---|---|---|---|---|---|---|---|---|---|---|
| AP | TP | Hard | Mask | Pan. | RAG | LAG | Eso. | Duo. | Liver | Pancreas | HepVes. | Colon | T4 |
| | | | | 78.18 | 69.63 | 69.08 | 76.99 | 54.39 | 66.37 | 42.05 | 62.20 | 39.85 | 51.17 |
| ✓ | | | | 83.55 | 72.80 | 71.65 | 79.31 | 60.45 | 64.82 | 45.08 | 68.72 | 43.84 | 53.91 |
| | ✓ | | | 80.62 | 71.02 | 70.34 | 72.81 | 57.31 | 69.13 | 44.31 | 65.18 | 40.16 | 52.32 |
| ✓ | ✓ | | | 83.50 | 72.71 | 69.96 | 78.99 | 64.08 | 72.49 | 44.55 | 69.40 | 44.50 | 55.84 |
| | ✓ | | ✓ | 86.74 | 72.41 | 69.00 | 77.68 | 59.99 | 69.12 | 43.23 | 67.75 | 41.32 | 54.33 |
| ✓ | ✓ | ✓ | | 87.36 | **74.46** | 74.02 | 75.39 | 70.80 | 72.64 | 48.49 | 69.02 | 47.29 | 53.67 |
| ✓ | ✓ | | ✓ | 88.49 | 73.24 | 74.51 | **80.76** | 70.26 | 72.18 | 46.46 | 69.97 | 46.65 | **58.49** |
| ✓ | ✓ | **✓** | ✓ | 88.28 | 74.42 | 72.50 | 79.30 | 71.26 | 70.95 | 45.52 | 69.51 | 46.07 | 56.41 |
| ✓ | ✓ | ✓ | ✓ | **89.24** | 73.69 | **74.63** | 80.10 | **73.46** | **72.73** | **49.67** | **70.11** | **48.31** | 57.37 |

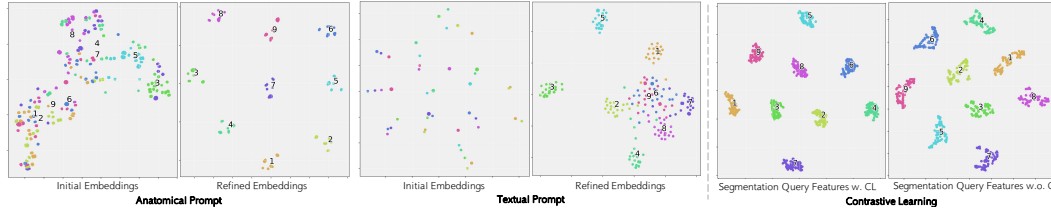

Figure 4: T-SNE visualization of the distribution of Features. Left: Two types of prompt embedding before and after refinement. Right: Segmentation query features with and without constrastive alignment. (1-9: right kidney, left kidney, liver, pancreas, colon, kidney tumor, liver tumor, pancreas tumor, colon tumor).

**Effectiveness of Module Designs.** As presented in the last five rows in Table 3, we evaluate the impacts of the module designs in our ShareRefiner and PromptRefer. The comparison of results between the seventh and last rows demonstrates that employing the hard assignment for refining the anatomical queries can boost performances, especially for tumor segmentation. However, applying the paradigm to the other two queries (i.e., the 'Hard' column is marked as ✓) will decrease the performance of segmentation. This decline can be attributed to the entanglement of the three query types. Such a phenomenon further reinforces that anatomical and textual prompts contribute from different perspectives. Using a one-hot hard assignment ensures that each query feature remains exclusive to the others, allowing queries to focus on distinct visual regions without overlap. We hypothesize that anatomical queries could attend to features and distinguish between different categories based on inherent appearance characteristics. Additionally, we remove the mask mechanism in the PromptRefer and utilize a vanilla cross-attention layer to update $\mathbf{Q}'_S$. As illustrated in Table 3, this variant leads to reduced performance in tumors that invade other organs (e.g., Stage-IV). The result verifies that guiding queries to focus on relevant objects is an effective strategy for achieving more robust segmentation in medical scenarios.

**Visualizations.** We select nine categories (five organs and four tumors) and use t-SNE [63] to visualize the distribution of prompt features ($\mathbf{E}_A$ and $\mathbf{E}_T$) and updated prompts queries ($\mathbf{Q}'_A$ and $\mathbf{Q}'_T$) in Figure 4. Our findings indicate that the initial prompt features are not well-separated in the feature space. For instance, the anatomical features of the right kidney and left kidney (categories 1 and 2) are entangled. Therefore, different refined paradigms need to be applied to update both queries simultaneously. With our ShareRefiner, prompt features are separated in the feature space. Moreover, we can observe that the distribution of refined anatomical prompt queries is more structured than textual prompts, which further verifies that the anatomical prompt is more intuitive. Contrastive alignment is utilized to further push segmentation queries to be close to the referenced prompt for segmenting the corresponding category. To validate the effectiveness, we trained without utilizing contrastive alignment. We also use t-SNE to visualize the distribution of decoded segmentation query features $\mathbf{O}_S$ in the right part of Figure 4. We can observe that segmentation queries are more separated in the feature space with contrastive alignment. Additionally, we visualize the heatmaps derived from experiments in the second and third rows of Table 3. As shown in Figure 5, solely using textual prompts fails to cover all regions, while only relying on anatomical prompts results in a high

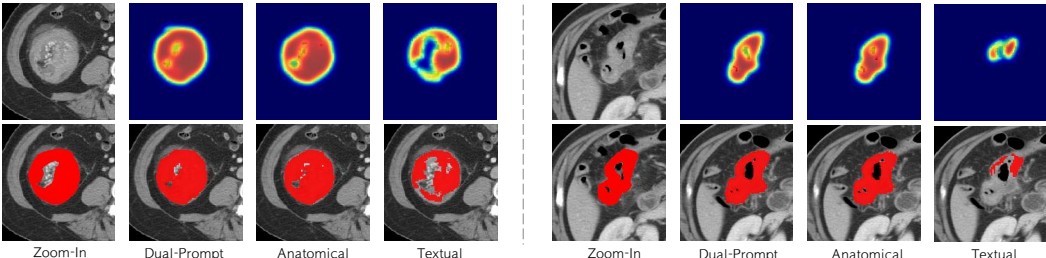

Figure 5: Heatmaps of two samples for analyzing the effectiveness of two prompts.

false positive rate. These visual examples underscore the importance of combining both types of prompts to guide medical image segmentation effectively.

## 5   Conclusion and Discussion

**Conclusion.** We present *CAT*, a promising attempt towards comprehensive medical segmentation via coordinating anatomical-textual prompts. Apart from performing generic organ segmentation, *CAT* can identify varying tumors without human interaction. To effectively integrate two prompt modalities into a single model, we design ShareRefiner to refine latent prompt queries with different strategies and introduce PromptRefer with specific attention masks to assign prompts to segmentation queries for mask prediction. Extensive experiments indicate that the coordination of these two prompt modalities yields competitive performance on organ and tumor segmentation benchmarks. Further studies revealed the robust generalization capabilities to segment tumors in different cancer stages. We hope our early exploration of the complementary advantages between anatomical and textual prompts could bring new insights into the field of the community.

**Limitation and Impact.** We hope our model can support professionals in the arduous clinical diagnosis process. Despite the integration of anatomical and textual prompts showing competitive performance in the segmentation tasks, the lack of general anatomical prompts encoders raises challenges, as indicated by the messy distribution of initial anatomical embeddings. Therefore, further research into improving the CT foundation models is essential. Moreover, there may also be mistakes in the segmentation results, especially when the test sample contains rare types of lesions or undergoes radical resection surgeries that cause large variations in the anatomical structures. Therefore, before integrating these AI-based algorithms into clinical practice, legislation needs to be developed and implemented to ensure that there are clear guidelines and standards for their use.

## Acknowledgments and Disclosure of Funding

This work was supported by the National Natural Science Foundation of China (No. 62301311) and the Shanghai Municipal Commission of Economy and Informatization (No. 204694).

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

# A   Appendix / supplemental material

## A.1   Textual Prompts Construction

| The instruction for GPT-4 | Short phrase templates |
|---|---|
| Please describe the provided organs and tumors. Descriptions should include the location, its anatomical structure, common pathological findings, and common tumors (for organs). Answers must be based on observations from CT imaging. | A computerized tomography of a [CLS]. A photo of a [CLS]. There is [CLS] in this computerized tomography. [CLS] |

Table 4: The strategy of constructing textual prompts.

We call GPT-4 to generate long descriptions for each category and use templates to construct some short textual prompts, as shown in Table A.1. We use the long description for the positive category and randomly sample short phrases for those negative categories. Positive categories indicate labeled categories in the case.

## A.2   Dataset Details

The overall categories used in our paper consist of 23 organs and 6 anomalies. The test set of MSD is the same as the official code of [13]. Details of utilized datasets are shown as follows:

(1) The BTCV dataset [50] consists of 30 abdominal CT scans from 13 annotated organs, conducted under the supervision of radiologists at Vanderbilt University Medical Center.

(2) The CT-ORG dataset [53] encompasses 140 CT images, featuring six organ classes. These images primarily display liver lesions, including both benign and malignant types.

(3) The AbdomenCT-1K dataset [54] comprises 1,112 CT scans pooled from five datasets, annotated for the liver, kidney, spleen, and pancreas.

(4) CHAOS [52] offers 40 CT scans of healthy abdominal organs for multi-organ segmentation, specifically excluding any pathological abnormalities such as tumors or metastasis.

(5) AMOS22 [51], a multi-modality abdominal multi-organ segmentation challenge of 2022, includes 500 CT scans with voxel-level annotations of 15 abdominal organs.

(6) The WORD dataset [56] collects 150 CT scans from patients prior to radiation therapy at a single center, with each volume consisting of 159 to 330 slices and comprehensively annotated for 16 anatomical organs.

(7) Pancreas-CT [55] contains 82 contrast-enhanced abdominal CT volumes focused solely on the pancreas, annotated by an experienced radiologist and excluding any pancreatic tumors.

(8) The LiTS dataset [57] comprises 201 contrast-enhanced abdominal CT scans (131 for training and 70 for testing), acquired across six clinical sites using various scanners and protocols, with a resolution range from 0.55 to 1.0 mm and slice spacing from 0.45 to 6.0 mm.

(9) KiTS [58] includes 300 CT scans (210 for training and 90 for testing) with annotations provided by the University of Minnesota Medical Center, each featuring one or more kidney tumors.

(10) The Medical Segmentation Decathlon (MSD) [59] comprises 947 CT scans targeting liver, lung, pancreas, colon, hepatic vessels, and spleen, encompassing a total of four organs and five tumors.

(11) Our in-house dataset contains 80 CT scans of patients diagnosed with colon cancer, annotated by an experienced gastroenterologist and verified by a senior radiologist. All scans maintain a consistent in-plane dimension of $512 \times 512$ pixels, while the z-axis dimension ranges from 36 to 146, with a median of 91. Note that there are some cases where the cancer stage can not be indentified.

## A.3   Implementation Details

AdamW optimizer [64] is utilized as the optimizer. The default initial learning rate is $1 \times 10^{-4}$, and we decay it following the learning rate scheduling strategy of [61]. Each volume is cropped into patches with a size of $96 \times 96 \times 96$. Random shift, zoom and scale, and scaling are applied on-the-fly to improve the generalization. Our framework is implemented using PyTorch and all experiments are

| Datasets | #Scans | Annotated categories |
|---|---|---|
| BTCV [50] | 30 | Spl, RKid, LKid, Gall, Eso, Liv, Sto, Aor, IVC, R&SVeins, Pan, RAG, LAG |
| CT-ORG [53] | 140 | Lung, Liver, Kidneys and Bladder |
| AbdomenCT-1K [54] | 1000 | Spleen, Kidney, Liver, Pancreas |
| CHAOS [52] | 40 | Liver, Left Kidney, Right Kidney, Spleen |
| AMOS22 [51] | 500 | Spl, RKid, LKid, Gall, Eso, Liv, Sto, Aor, IVC, Pan, RAG, LAG, Duo, Bla, Pro/UTE |
| WORD [56] | 150 | Spl, RKid, LKid, Gall, Eso, Liv, Sto, Pan, RAG, Duo, Col, Int, Rec, Bla, LFH, RFH |
| Pancreas-CT [55] | 82 | Pancreas |
| LiTS [57] | 201 | Liver, *Liver Tumor* |
| KiTS [58] | 300 | Kidney, *Kidney Tumor* |
| MSD CT Tasks [59] | 947 | Spl, Liver, *Liver Tumor*, *Lung Tumor*, *Colon Tumor*, Pancreas and *Pancreas Tumor*, Hepatic Vessel and *Hepatic Vessel Tumor* |
| In-house dataset | 80 | *Colon Tumor* with four subtypes |

Table 5: The information for all datasets.

conducted on 8 NVIDIA A100 GPUs. For the evaluation, the Dice Similarity Coefficient (DSC) and 95% Hausdorff Distance (HD95) are utilized to gauge the performance of organ/tumor segmentation.

## A.4 Results Details

Table 6: Organ segmentation performance on FLARE22. The results(%) are evaluated by DSC. Scores of SAM-based are adopted from the CT-SAM-Med3D [41]. † denotes obtained via the official pre-trained weights. ∗ means implemented from the official code and trained on the same dataset. Abbreviations: "Liv."-Liver, "R_Kid."-Right Kidney, "Spl."-Spleen, "Pan."-Pancreas, "Aor."-Aorta, "IVC"-Inferior Vena Cava, "RAG"-Right Adrenal Gland, "LAG"-Left Adrenal Gland, "Gal."-Gallbladder, "Eso."-Esophagus, "Sto."-Stomach, "Duo."-Duodenum, " L_Kid."-Inferior Vena Cava.

| Methods | Liv. | R_Kid. | Spl. | Pan. | Aor. | IVC | RAG | LAG | Gal. | Eso. | Sto. | Duo. | L_Kid. |
|---|---|---|---|---|---|---|---|---|---|---|---|---|---|
| SAM [22] | 86.0±5.5 | 87.6±8.7 | 84.5±8.9 | 53.4±10.6 | 77.5±16.1 | 44.5±16.3 | 19.4±14.0 | 33.9±15.0 | 52.4±16.4 | 35.2±6.8 | 68.0±11.2 | 44.4±12.4 | 82.6±11.3 |
| MedSAM [17] | 93.0±3.1 | 90.0±5.3 | 89.1±11.0 | 73.5±9.6 | 82.5±19.7 | 76.5±19.4 | 36.0±23.6 | 48.7±22.6 | 56.4±27.1 | 64.7±19.9 | 84.0±13.0 | 53.9±11.7 | 89.7±7.9 |
| SAM-Med2D [19] | 91.4±5.8 | 83.7±17.3 | 83.9±15.2 | 58.8±18.7 | 60.6±22.5 | 18.6±10.4 | 10.6±9.7 | 27.1±12.4 | 32.9±21.6 | 28.1±13.3 | 72.9±16.6 | 45.4±19.8 | 86.0±16.8 |
| SAM-Med3D [40] | 85.4±13.2 | 84.2±9.5 | 84.7±11.8 | 46.9±14.3 | 60.4±10.7 | 44.5±13.4 | 32.6±20.9 | 35.3±18.3 | 56.0±19.4 | 32.6±16.4 | 46.9±19.8 | 27.4±13.6 | 84.9±6.9 |
| SegVol [39] | 83.9±25.3 | 71.7±30.6 | 75.9±28.8 | 69.4±16.1 | 83.1±12.1 | 80.3±13.9 | 42.1±13.3 | 49.7±22.6 | 55.6±31.1 | 69.6±8.4 | 81.1±20.7 | 55.6±19.8 | 75.1±22.6 |
| CT-SAM3D [41] | 95.6±2.0 | 95.0±1.8 | 96.1±4.4 | 83.6±12.0 | 94.5±2.8 | 91.8±4.7 | 78.4±18.0 | 82.5±4.0 | 88.4±8.1 | 82.9±18.1 | 92.3±4.4 | 73.2±16.8 | 94.8±1.4 |
| Universal† [13] | 97.4±1.4 | 95.5±6.7 | 96.4±1.45 | 73.7±29.6 | 84.9±10.2 | 84.4±9.4 | 72.9±19.5 | 73.4±14.3 | 86.0±8.8 | 76.8±17.4 | 88.5±16.0 | 74.5±11.3 | 96.9±0.8 |
| ZePT* [12] | 96.7±2.0 | 95.6±7.7 | 96.6±2.9 | 84.3±2.5 | 90.0±9.0 | 84.4±2.5 | 67.2±20.4 | 66.8±10.1 | 79.6±9.5 | 74.2±17.3 | 85.2±11.1 | 59.1±11.8 | 97.2±1.0 |
| CAT | **97.7±1.3** | **96.3±6.1** | **97.1±1.1** | **89.2±7.6** | 90.5±9.1 | 88.0±2.9 | 73.6±18.3 | 74.3±10.0 | 83.0±10.0 | 80.1±9.2 | 88.2±12.3 | 73.4±8.8 | **97.3±0.7** |

Table 7: Segmentation performance of tumors on MSD [59] and In-house dataset. We compare our method with traditional and promptable methods. † denotes obtained via the official pre-trained weights. ∗ means implemented from the official code and trained on the same dataset.

| Method | MSD Dataset (Tumor in Abdomen) | | | | In-house Data (Colon Tumor) | | | | |
|---|---|---|---|---|---|---|---|---|---|
| | Liver | Pancreas | Hepatic Vessel | Colon | T1 | T2 | T3 | T4 | Avg. |
| nnUNet* [62] | 66.42±26.5 | 43.50±34.1 | 66.90±21.1 | 41.41±27.8 | 19.51±33.8 | 45.06±33.9 | 44.87±32.4 | 45.54±29.9 | 43.00±31.9 |
| Swin UNETR* [48] | 68.67±23.1 | 41.77±32.9 | 63.32±25.5 | 39.35±28.6 | 21.40±22.2 | 33.32±30.4 | 46.11±25.3 | 52.72±22.8 | 45.92±26.5 |
| SAM-Med3D† [40] | 44.78±38.7 | 40.05 ±18.3 | 44.86±26.3 | 39.23±19.1 | 34.28±23.4 | 42.65±19.1 | 50.20±16.9 | 42.65±18.8 | 47.11±19.2 |
| SegVol† [39] | 66.20±26.0 | 46.36±28.8 | 68.57±28.2 | **60.63±24.1** | **36.93±33.5** | 42.63±27.8 | **60.17±27.2** | 49.83±28.1 | 50.28±28.7 |
| Universal† [13] | 65.68±27.2 | 45.72±31.9 | 66.31±21.5 | 42.26±30.4 | 7.11±12.3 | 43.28±29.6 | 46.52±28.6 | 53.08±26.9 | 47.14±29.1 |
| ZePT* [12] | 68.58±24.3 | 44.39±34.2 | 68.12±20.6 | 40.38±29.9 | 23.87±23.8 | 34.64±32.2 | 50.81±28.2 | 51.09±25.5 | 46.28±28.5 |
| CAT | **72.73±23.3** | **49.67±32.7** | **70.11±19.7** | 48.31±27.3 | 30.62±31.5 | **45.61±30.8** | 55.85±27.7 | **57.37±23.4** | **53.35±27.0** |

Further results are depicted in Figure 3. Our model demonstrates the capability to segment all structures of the aorta, including the arcus aortae. This comprehensive segmentation is advantageous in real-world scenarios, although it slightly lowers the Dice score.

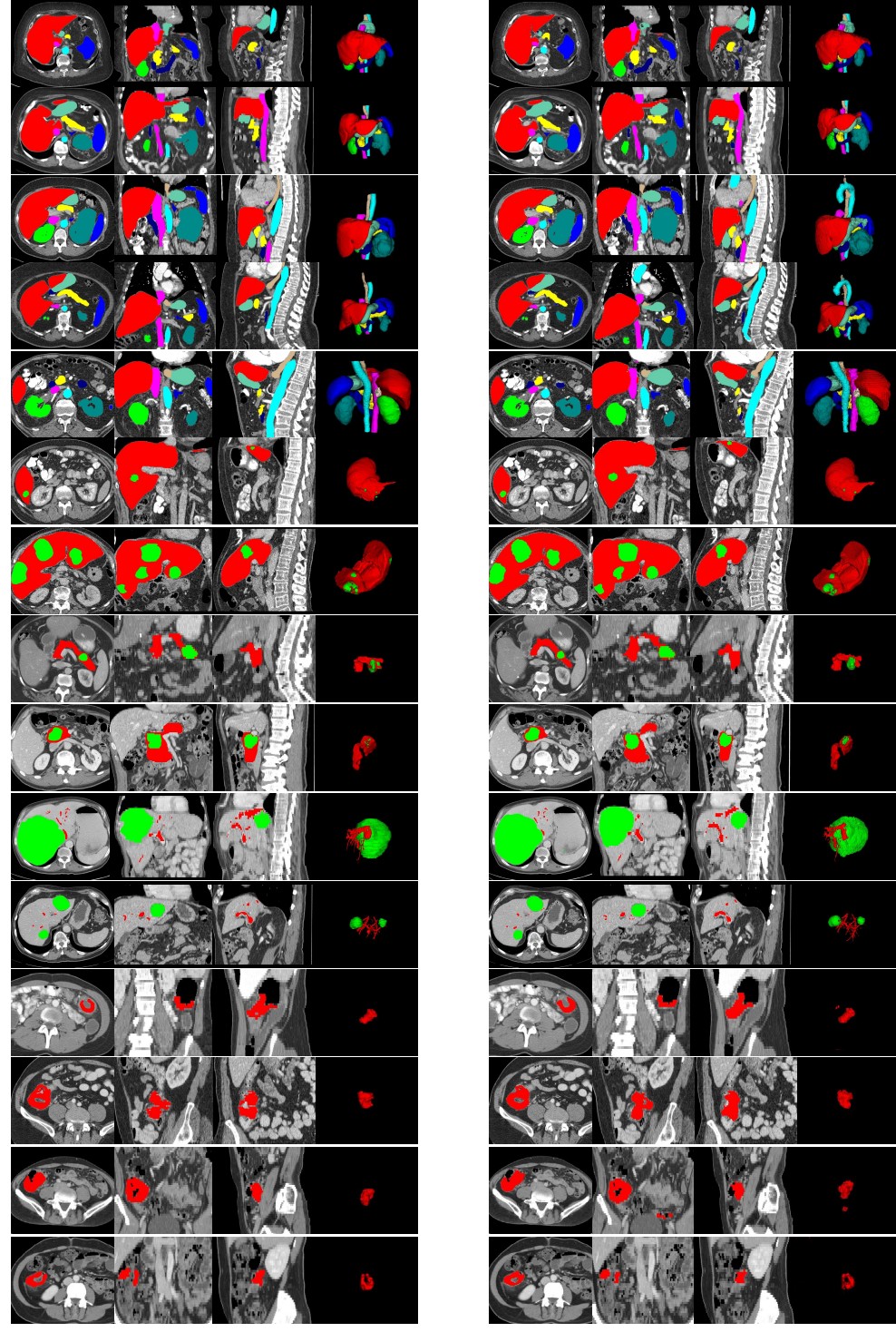

Figure 6: Visualization of segmentation result of **CAT**. The left part is the ground truth and the right part is ours. The first five rows are results of organ segmentation results, and the others are tumor segmentation results. From the top to down are liver tumors, pancreas tumors, Hepatic Vessel tumors, colon tumors, and colon tumors in T4. For each tumor type, we present two cases.

