# OpenReview forum: "CAT: Coordinating Anatomical-Textual Prompts for Multi-Organ and Tumor Segmentation"
_NeurIPS.cc/2024/Conference — NeurIPS 2024 poster_

### Official Review · Reviewer_Yp5t · 2024-07-03

**Soundness:** 3
**Presentation:** 3
**Contribution:** 3
**Rating:** 5
**Confidence:** 5

**Summary:**

This paper proposes a prompt-based deep model for organ and tumor segmentation. The authors leverage two types of prompts—cropped target volumes and textual descriptions—to perform the segmentation. The proposed model demonstrates good performance across three datasets for the segmentation task.

**Strengths:**

1. Point out several critical challenges in medical image segmentation, such as the long-tailed nature, variations in shape, size, and density distribution, and blurring boundaries.
2. Utilize both textual and visual prompts for medical image segmentation, achieving good performance.

**Weaknesses:**

1. The motivation is not convincing. Although the authors highlight several critical issues, it remains unclear how the two types of prompts address or refine them. For instance, textual descriptions lack detailed and quantitative measurements. How can they denote invading target boundaries? How can the effectiveness of terms like 'greater' or 'deeper' be evaluated?

2. Several experimental settings are not standard. For example, in the Anatomical Prompt Encoder, the input sizes are the same. How does this account for small or large targets?

3. Figure 2 is unclear. What is the output? During inference, do both prompt inputs need to be used?

4. For comparisons, the authors trained the model on 10 public datasets. However, only ZePT, nnUNet, and Swin UNETR were trained in the same setting. Therefore, the conclusions drawn are not reasonable. Besides, in Table 1, CT-SAM3D achieves the best performance.

5. The overall organization is unsatisfactory. First, as mentioned, the motivations could be improved to better align with the technical designs. Second, despite the better performance, no new insights are provided.

**Questions:**

See the above weaknesses.

**Limitations:**

The authors have analyzed the limitations of this paper.

---

> ### Author Rebuttal · Authors · 2024-08-05
>
> Thank you very much for your constructive suggestions. Below is our detailed response to answer your concerns.
> # W1&W5: Motivation and Details of textual prompts
> The motivation for using textual descriptions was to provide the model with the general concept of each category. While descriptive texts encompass intricate and rare anomalies using domain knowledge, learning alignments between textual and visual representations remains challenging, particularly for fine-grained details such as tumors with variations in shape, size, density distribution, and blurred boundaries. In real-world scenarios, acquiring specific details like quantitative measurements for each sample prior to segmentation is often impractical. Considering the vast diversity of tumors, we utilize general knowledge to convey the concept of each tumor type. Given the vast diversity of tumors, we employ general knowledge to convey each tumor type's concept. Utilizing GPT-4, we generated descriptions with medical domain knowledge for each category over 20 times. A board-certified physician then refined these descriptions based on the results. Further details on our textual prompts are discussed in the **[Text Descriptions](https://anonymous.4open.science/r/Reb/textual_prompts.json)**. For example, the description of colon tumors ("On CT scans, colon tumors can present with varying densities and might be associated with surrounding inflammation, adjacent organ invasion, or regional lymph node enlargement.") contains an abstract notion of invading target boundaries.  During the training process, we use the long description for the positive categories and randomly sample short phrases for those negative categories. In the inference stage, we apply the long description for all categories.  To verify the effectiveness of textual descriptions, we replace them with short phrases during the inference stage, the result can be seen as follows:
>
> |Dice(%)|Pan.|RAG.|LAG.|Eso.|Duo.|Liv. Tumor|Pan. Tumor|HV. Tumor|Colon Tumor|Colon Tumor(T4)|
> |:-------------------|:------:|:------:|:------:|:-------:|:------:|:--------------:|:-----------------:|:-----------------------:|:--------------:|:-------------------:|
> |CAT w. short phrases|88.87|72.17|73.12|74.37|68.99|    70.65|47.80|68.77|46.31|54.09|
> |CAT|89.24|73.69|74.63|80.10|73.46|72.73|49.67|70.11|48.31|57.37|
>
> The observed declines demonstrate that textual descriptions, which cover the general concept of potential cases, significantly benefit the tumor segmentation process. To provide an intuitive understanding of these results, we present the qualitative results in the **Figure** (Figure 6 of the rebuttal PDF). Additionally, we enlisted a physician to annotate the tumor regions. The figure shows that a lack of detailed knowledge leads to overlooking crucial details. Importantly, the results from CAT are more closely aligned with those delineated by the expert, who possesses professional medical knowledge. This validates our approach of enhancing the segmentation process by incorporating comprehensive textual knowledge from the medical domain.
>
> We hope the updated clarification in the **Author Rebuttal** could provide a clear understanding for you.
>
> # W2:  Details of Anatomical Prompt
> We are very grateful for your serious reading. In our paper, we leverage the bounding box derived from masks in the public dataset and relevant anatomical structures to prepare a set of prompt volumes for each category, which we then standardize to a uniform size. The process begins by cropping the image according to the bounding box. Subsequently, we employ two different strategies to adjust the cropped volumes to the required input size of the Anatomical Prompt Encoder ($96 \times 96 \times 96$): 1. For volumes larger than the target size, we resize them to $96 \times 96 \times 96$. 2. For smaller volumes, we re-crop the image using a center-crop paradigm to achieve the size of $96 \times 96 \times 96$. It is important to note that all anatomical prompts used for tumor categories are processed using the second strategy. We exclude cases where the tumor size exceeds $96 \times 96 \times 96$, as these instances have been effectively addressed by previous methodologies. Our paper primarily focuses on the integration of anatomical and textual prompts for challenging segmentation tasks.
> # W3: Clarification of Figure 2
> Sorry for any misunderstanding caused by the omission of image descriptions. As shown in the **Figure** (Figure 7 of the rebuttal PDF), the part highlighted by a red box is the output. We chose this method of illustration to provide a more intuitive understanding; we regret any confusion this may have caused. The segmentation maps are obtained by a multiplication operation between the decoded segmentation query features $\mathbf{O}_S$ and the pixel embedding map $O$. During inference, both prompt inputs need to be used.
> # W4: Experimental Details
> Sorry for any confusion caused by missing details in our experimental descriptions. In our experiments, the results for the Universal model were derived from the official pre-trained weights, which were also trained on the same datasets as our study. In contrast, the SAM-based models were trained on a broader set of datasets, including the ten datasets discussed in our paper.  Regarding the CT-SAM3D model, the detailed organ-wise segmentation results were sourced from the original CT-SAM-Med3D paper. It is important to note that the results presented were obtained using a prompt number ($N=5$), which is derived from the ground truth. As indicated in the subsequent **Figure** (Figure 3 of the rebuttal PDF), there is a significant performance drop in the average Dice score when the number of prompts decreases from 5 to 1, falling below 85%. Thanks to the design of our CAT model, it demonstrates superior results in organ segmentation without the need for human interaction

---

> > ### Comment · Reviewer_Yp5t · 2024-08-10
> > **Thank you for the reply**
> >
> > Thanks for the response and it solves most of my concerns. Using both text and visual prompts is a kind of combination and it is not surprising it could achieve better performance. Even though the technical novelty is not strong, the analysis of clinical usage is reasonable and I will raise my score to borderline acceptance. This is the final comment.

---

> > > ### Author Response · Authors · 2024-08-10
> > >
> > > We sincerely appreciate your thorough review and constructive feedback on our manuscript. Your detailed comments have provided invaluable insights that have significantly contributed to the refinement of our work.

---

### Official Review · Reviewer_qosx · 2024-07-03

**Soundness:** 3
**Presentation:** 3
**Contribution:** 2
**Rating:** 5
**Confidence:** 4

**Summary:**

The paper "CAT: Coordinating Anatomical-Textual Prompts for Multi-Organ and Tumor Segmentation" introduces a novel dual-prompt schema that leverages both anatomical and textual prompts for medical image segmentation. The proposed CAT model coordinates 3D anatomical prompts with enriched textual prompts to improve segmentation accuracy for various organs and tumors. Key contributions include the development of the ShareRefiner and PromptRefer modules to refine and integrate these prompts, resulting in superior performance on multiple public CT datasets and an in-house dataset. The approach demonstrates enhanced generalization capabilities and robustness in complex medical imaging scenarios.

**Strengths:**

1. The paper introduces a novel approach by combining anatomical and textual prompts, leveraging the strengths of both to enhance medical image segmentation.

2. The development of the ShareRefiner and PromptRefer modules demonstrates a sophisticated method for refining and integrating multimodal prompts, leading to improved segmentation accuracy.

3. The paper provides a thorough experimental analysis, including ablation studies and qualitative comparisons, to substantiate the effectiveness of the proposed methods.

4. The work focuses on both organ and tumor segmentation, which is commendable and proves the model's performance overall.

**Weaknesses:**

1. The motivation for using textual descriptions was to provide the model with specific knowledge about each image, including details such as cancer stages and density. However, the authors use only fixed generic textual information generated by a language model in this work. This approach does not fully capture the intended motivation. If the textual information included specific details for each sample, demonstrating how changes in staging affect segmentation performance, it would better support their claim about the utility of textual information.

2. I am not convinced of CAT's superiority in the comparative results. For example, CT-SAM3D seems to perform better in organ segmentation in most cases, and where CAT excels, the improvement is minimal compared to other text-based segmentation models. Additionally, why aren’t SOTA segmentation models like UNet, nn-UNet, and Swin UNETR, etc included for organ segmentation, and vice versa for tumor segmentation? Only comparing with SAM-based models for organ segmentation is insufficient, especially since SAM is known to have poor performance in medical image segmentation.

3. The authors claim that CAT performs better on the in-house dataset, especially for tumors at different stages. However, the textual prompts used by CAT are very generic and do not include the cancer stage information. Thus, the reasons for CAT's superior performance are unclear.

4. The paper lacks sufficient novelty, as it combines already existing methods, such as integrating text and visual prompts, where text is intended to provide rich semantic information. However, it does not adequately explain how the text contributes to performance improvement. This work appears to be an incremental extension of ZePT, with only the creation of visual prompts distinguishing it from similar efforts.

5. Exactly how contrastive alignment works in PromptRefer isn't clear. This should be further enhanced.

6.  Many of the techniques are described in terms of their usage, but the underlying motivation for their utilization is not clearly articulated. For instance, the rationale for choosing hard assignments in ShareRefiner is not explained. If the intention is to follow ZePT, this choice is questionable because ZePT uses hard assignments to distinguish between healthy organs and tumors in feature space. The motivation behind these decisions is lacking.

7. Figure 2 (c) should have more explanations. It is not clear right now.

Overall, the authors should place greater emphasis on the motivations behind the chosen techniques. Currently, it appears they are following these methods simply because they work, which is not a robust standard. Clear justification for each technique would strengthen the paper.

**Questions:**

1. In equation 3, why is a Linear function used to convert EAE_AEA​ to QAQ_AQA​? There is no explanation provided for this choice. I assume it is to reduce the dimensions of the embedding space, but clarification is needed.

2. There should be more explanations about learnable segmentation queries. For example, how are they initialized? What is their purpose? Beyond their use in architecture, a high-level explanation of how they help predict masks would aid readers in understanding and relating to the motivation behind these techniques.

3. Why do some models present HD95 values while others do not? The current reasoning isn't sufficient.

4. Figure 1, some modules have the "snowflake" icons, which the authors don't explain why. Assuming these show that these are frozen models that have not been trained, do they mean any modules that don't have these are all trained?

**Limitations:**

Please refer to the Weaknesses section.

---

> ### Author Rebuttal · Authors · 2024-08-05
>
> Thank you for your thorough review. Below, we will address your concerns on each point.
> # W1&W3: Details of textual descriptions and Reasons for superior performance
> As detailed in Section 3.4, for each category, we curate long descriptions. As highlighted in the motivation section, learning the alignments between textual and visual representations is challenging, particularly for fine-grained details. In real-world scenarios, obtaining specific details for each sample prior to segmentation is impractical. Considering the vast diversity of tumors, we leverage general knowledge to convey the concept of each tumor type. Details can be in **[Textual Prompts](https://anonymous.4open.science/r/Reb/textual_prompts.json)**. To better support our claim about the utility of textual prompts, we conducted ablation studies in Table 3 of our paper. The observed improvement in the third row (compared to the first row) underscores the effectiveness of textual prompts in helping deal with the diversity and variance of tumors.
>
> As shown in Table 2, CAT shows strong capabilities in dealing with tumors at different stages. The reasons can be attributed to several factors: 1. The textual prompt for colon tumors provides context for '**surrounding inflammation, adjacent organ invasion, or regional lymph node enlargement**,' which are critical in different stages. 2. The visual prompts offer intuitive and direct examples to highlight the appearance features.  3. The carefully designed mask in the PromptRefer helps to handle tumors that invade nearby organs or tissues.
> # W2: Organ segmentation results
> 1. The detailed organ-wise segmentation results are derived from the original CT-SAM-Med3D paper. It is important to note that the results presented were achieved with a prompt number ($N=5$), which is derived from the ground truth. As illustrated in the cited **Figure** (Figure 3 of the rebuttal PDF) from CT-SAM-Med3D, reducing the number of prompts from 5 to 1 leads to a significant performance decrease, with the average Dice score falling below 85%.
>
> 2. The average organ segmentation scores for nn-UNet and Swin UNETR are presented below. Our design enables CAT to outperform most SAM-based models in the medical domain and established state-of-the-art (SOTA) segmentation models.
>
> - **nn-UNet**: Pan.-79.36, RAG.-68.37, LAG.-73.31, Eso.-76.16, Duo.-72.43, **Avg.-83.38**
> - **Swin UNETR**: Pan.-87.05, RAG.-66.17, LAG.-74.18, Eso.-75.65, Duo.-48.51, **Avg.-82.78**
>
> # W4&W6&W7: Details of our method
> Factually, CAT significantly diverges from ZePT in several key aspects. Besides the introduction of visual prompts, CAT utilizes more descriptive texts as textual prompts to convey general concepts, while ZePT only uses knowledge for the final alignment. For handling two types of prompts, CAT employs distinct strategies: we use soft assignment to gather all potentially relevant visual features for textual prompts while applying hard assignment to secure discriminative visual regions without overlaps.  This is markedly different from ZePT’s unified refinement process. Furthermore, in PromptRefer, we carefully design attention masks according to clinical knowledge. In summary, CAT mainly focuses on combining anatomical and textual prompts to enhance segmentation tasks, whereas ZePT delves into identifying anomalies from original visual features.
>
> Figure 2(c) illustrates how segmentation queries interact with a mixed group of refined prompt queries via a cross-attention mechanism in the PromptRefer module. For example, Stage-IV colon tumors often invade adjacent organs such as the intestines. In such cases, the segmentation query $\mathbf{Q'}_{Si}$ for colon tumors is specifically directed to attend to the refined query set of anatomical $\mathbf{Q'}_A$ and $\mathbf{Q'}_T$ encompassing Colon, Stomach, Duodenum, Intestine, and Rectum. We hope the **Figure** (Figure 4 of the rebuttal PDF) clarifies you more clearly.
> # W5: Effectiveness of contrastive alignment
> Contrastive alignment is utilized to further push segmentation queries to be close to the referenced prompt for segmenting the corresponding category. To validate the effectiveness, we trained without utilizing contrastive alignment. The results are shown in the following table. Eliminating contrastive alignment leads to a performance drop.
>
> |Dice(%)|Pan.|RAG.|LAG.|Eso.|Duo.|Liv. Tumor|Pan. Tumor|HV. Tumor|Colon Tumor|Colon Tumor(T4) |
> |:-----------------------------|:------:|:------:|:------:|:-------:|:------:|:--------------:|:-----------------:|:-----------------------:|:--------------:|:-------------------:|
> |CAT w/o  contrastive alignment|87.97|73.61|72.45|79.06|71.35|71.36|47.61|68.10|46.50|56.07|
> |CAT|89.24|73.69|74.63|80.10|73.46|72.73|49.67|70.11|48.31|57.37|
>
> We also use t-SNE to visualize the distribution of decoded segmentation query features $\mathbf{O}_S$ in **Figure** (Figure 5 of the rebuttal PDF).  We can observe that segmentation queries are more separated in the feature space with contrastive alignment.
> # Questions:
> Thanks for your careful reading. As you mentioned, the Linear function is to transform the dimensions of the embedding space. The "snowflake" icons in Figure 2 indicate that these are frozen models that have not been trained. Following previous work, the learnable segmentation queries are initialized randomly. Each of them is responsible for segmenting one category.
>
> The reason for not reporting HD95 scores for SAM-based methods: The HD95 metric is used to assess the accuracy of image segmentation by measuring the largest distances between predicted and actual segmentations. When applying SAM-based methods that require predefined target key points as the prompts for segmentation, the predefined keypoints could artificially enhance segmentation accuracy near those points. Hence, reporting HD95 scores for SAM-based methods against others without such provisions is considered inequitable.

---

> ### Comment · Reviewer_qosx · 2024-08-11
> **Excellent Rebuttals.**
>
> Thanks for the response. I still have some doubts:
>
> * I understand that adding generic definitions has led to some improvement, but it doesn't fully align with the motivation of your work. For example, it's unclear if your text encoder can effectively differentiate between different cancer stages in real-life scenarios. Given that your paper focuses on the challenges of long-tail distribution (typically for tumors), this should have been demonstrated experimentally rather than assumed. If your paper's motivation was simply "Do organ definitions improve organ segmentation performance?" then the approach would be more acceptable.
>
> * Regarding the results, shouldn't all the organs be reflected in Table 1? The current comparison seems limited to only a few organs, which can be confusing. A more comprehensive comparison would be helpful. Also, outperforming SAM-based models alone isn't sufficient since it's well-known that SAM models don't excel in medical imaging tasks.
>
> Thank you for clarifying the other points. Including these explanations in the original paper would greatly enhance its clarity. Additionally, consider adding the contrastive alignment results to your ablation studies for a more thorough analysis.

---

> > ### Author Response · Authors · 2024-08-12
> >
> > We sincerely appreciate your thorough review and constructive feedback on our manuscript.  We are lucky to have met such a rigorous reviewer like you.  We will clarify your doubts and address your concerns on each point.
> >
> > ## Motivation
> >
> > Sorry for the earlier confusion regarding our motivation. The reason of introducing textual descriptions was to furnish the model with a general concept of each category.  As established in previous work [1] [2], both definitions of organs and tumors (i.e., textual prompts) can enhance segmentation results. Therefore, it is reasonable to guide the segmentation process via the texutal prompts. However, text-guided segmentation requires effective alignment between textual and visual representations. For instance, to segment a colon tumor in T-stage 4, descriptive texts encompassing all intricate and rare cases must be provided, and the deep learning model needs to determine which situation described in the textual description corresponds to the given visual sample. In the medical domain, aligning specific details with visual information is impractical due to the presence of numerous corner cases, such as tumors with variations in shape, size, density distribution, and blurred boundaries. Furthermore, accurately determining the T-stage in the TNM staging system involves multiple modalities, including image observation (MRI, CT, PET), physical examinations, and microscopic examination of biopsy samples, necessitating more comprehensive descriptions. Unfortunately, current text encoders struggle to differentiate effectively between different cancer stages when faced with lengthy descriptions.  Therefore, we utilize general knowledge to convey the concept of each tumor type instead of providing cumbersome descriptions for each T-stage. Our prompt,  '**surrounding inflammation, adjacent organ invasion, or regional lymph node enlargement**,'  could provide a general concept of colon tumors in different stages.  To verify the effectiveness of these textual descriptions, we replaced them with short phrases (e.g., "a CT image of a colon tumor") during the inference stage. The results can be observed as follows:
> >
> > | Dice(%)              | Pan.  | RAG.  | LAG.  | Eso.  | Duo.  | Liv. Tumor | Pan. Tumor | HV. Tumor | Colon Tumor | Colon Tumor(T4) |
> > | :------------------- | :---- | :---- | :---- | :---- | :---- | :--------- | :--------- | :-------- | :---------- | :-------------- |
> > | CAT w. short phrases | 88.87 | 72.17 | 73.12 | 74.37 | 68.99 | 70.65      | 47.80      | 68.77     | 46.31       | 54.09           |
> > | CAT                  | 89.24 | 73.69 | 74.63 | 80.10 | 73.46 | 72.73      | 49.67      | 70.11     | 48.31       | 57.37           |
> >
> > The observed declines demonstrate that textual descriptions, which cover the general concept of potential cases, significantly benefit the tumor segmentation process. To provide an intuitive understanding of these results, we present the qualitative results in the **Figure** (Figure 6 of the rebuttal PDF). Additionally, we enlisted a physician to annotate the tumor regions. The figure shows that a lack of detailed knowledge leads to overlooking crucial details. Importantly, the results from CAT are more closely aligned with those delineated by the expert, who possesses professional medical knowledge. This validates our approach incorporating such textual knowledge from the medical domain.
> >
> > To assess CAT’s efficacy in dealing with rare cases (i.e., the long-tailed problem), we introduce an in-house dataset where colon tumors have invaded adjacent organs. According to medical literature [3] [4], T4 colorectal tumors, which represent only about 5-8.8% of colon tumor cases, pose significant challenges in diagnosis and treatment. Our results demonstrate that CAT significantly outperforms other models in segmenting T4 colon tumors, underscoring the effectiveness of our design in handling complex medical scenarios.
> >
> > [1] Clip-driven universal model for organ segmentation and tumor detection.
> >
> > [2] ZePT: Zero-Shot Pan-Tumor Segmentation via Query-Disentangling and Self-Prompting
> >
> > [3] Results after multi-visceral resections of locally advanced colorectal cancers: an analysis on clinical and pathological t4 tumors.
> >
> > [4] Identification of risk factors for lymph node metastasis of colorectal cancer.

---

> > > ### Author Response · Authors · 2024-08-12
> > >
> > > ## Organ segmentation results
> > >
> > > Thank you for your suggestions. In our experiments, we primarily focus on organ segmentation in the abdomen. Therefore, we utilize FLARE22 as our test set, which includes 13 abdominal organs and is widely used to evaluate performance in the organ segmentation task. To further verify the effectiveness of our approach, we compare our results not only with SAM-based models but also with state-of-the-art models like Universal, which has demonstrated notable performance in organ segmentation. Our CAT model demonstrates superior results compared to state-of-the-art (SOTA) models. For a more comprehensive comparison, we present the results for additional organs in the table below; the experiments are conducted on the test set of our assembled dataset. All the models are trained on the same datasets. Our model demonstrates notable performance.
> > >
> > > | Dice(%)    | Colon | Intestine | Rectum | Prostate/Uterus | Bladder | Left Head of Femur | Right Head of Femur |
> > > | :--------- | ----- | :-------: | :----: | :-------------: | ------- | :----------------: | :-----------------: |
> > > | nn-UNet    | 69.20 |   76.76   | 71.38  |      73.38      | 84.37   |       88.27        |        88.25        |
> > > | Swin UNETR | 69.79 |   77.22   | 69.61  |      71.49      | 85.94   |       88.46        |        88.52        |
> > > | Universal  | 72.37 |   78.98   | 73.72  |      74.05      | 86.66   |       89.65        |        90.15        |
> > > | ZePT       | 70.41 |   75.64   | 72.74  |      77.76      | 86.91   |       90.58        |        90.45        |
> > > | CAT        | 72.61 |   79.95   | 74.03  |      78.82      | 87.71   |       91.09        |        91.86        |
> > >
> > > Given that the core objective of medical segmentation is to segment anomalies, our model primarily focuses on identifying varying tumors autonomously by coordinating anatomical and textual prompts. We hope our early exploration will bring new insights to the community and support professionals in the arduous clinical diagnosis process. Your detailed comments have provided invaluable insights that have significantly contributed to the refinement of our work.

---

> > > ### Comment · Reviewer_qosx · 2024-08-12
> > > **Final Comment**
> > >
> > > Thank you for the explanation. This makes sense.
> > > Most of my concerns have been addressed, I will be updating my score accordingly.

---

> > > > ### Author Response · Authors · 2024-08-12
> > > >
> > > > Thank you very much for reviewing our paper. Your constructive suggestions are valuable to us.

---

### Official Review · Reviewer_wEKt · 2024-07-13

**Soundness:** 3
**Presentation:** 3
**Contribution:** 3
**Rating:** 6
**Confidence:** 3

**Summary:**

This paper proposed CAT, a promptable segmentation model that utilizes the strengths of both visual and textual prompts without human interaction, aiming at a fully automatic model for medical professionals. Extensive experiments demonstrate the benefits of coordinating anatomical prompts and textual within one model. CAT achieves state-of-the-art performance on multiple segmentation tasks and has generalization capability to diverse tumor types.

**Strengths:**

1. The idea is OK. CAT combines text and visual prompts, which could be needed in clinical scenarios.
2. The experiment CAT proves that combining visual and textual prompts is essential.
3. The experiment shows that CAT can deal with challenging small region segmentation and tumor segmentation.
4. CAT applied domain knowledge generated from GPT4, which is innovative. And one board-certified physician is recruited to check the text prompts.

**Weaknesses:**

1. It is not clear how those comparison methods were trained. Were they also trained on the same 10 datasets as CAT? For example, how was nnUNet trained and tested (since the number output channel of nnUNet is fixed)?
2. The writing is a bit confusing. For example, what is the backbone of ShareRefiner and PromptRefer? The similarity matrices need some mathematical illustration.
3. It is confusing that Table 3 has the same setting in the last two rows. Is that a typo?
4. The paper did not mention what pre-trained model is used.

**Questions:**

1. What do the textual prompt and visual prompt look like? It is suggested that an example be given in the Supplementary.
If CAM is only trained on one dataset, will it surpass nnUNet and other comparisons?
2. How large are CAT and those comparison models? It is suggested to report the number of parameters.
3. It is suggested to give a concrete example that how visual and text prompt improve the segmentation.

**Limitations:**

The paper proposed in the introduction that medical data is challenging because of the long-trailed problem but did not illustrate how CAT helps solve this problem.

---

> ### Author Rebuttal · Authors · 2024-08-05
>
> Thank you for your constructive comments. We will address your concerns in the following parts.
> # W1: Details of Experiments
> Sorry for the confusion caused by the omission of certain experimental details. In our experiments, we followed Universal's experimental settings [1]. We also trained the comparison methods that we implemented ourselves on the same 10 datasets. Specifically, we modified the number of output channels in the baseline models (e.g., nnUNet, Swin UNETR) to enable them to perform multi-organ and tumor segmentation.
> # W2: Details of ShareRefiner and PromptRefer
> We hope the provided illustration will give you a clearer understanding of our ShareRefiner and PromptRefer. Both ShareRefiner and PromptRefer are built upon the attention mechanism [2].  Specifically, the ShareRefiner consists of a series of cross-attention blocks where each type of query (i.e., segmentation queries $\mathbf{Q}_S$ , anatomical prompt queries $\mathbf{Q}_A$ and textual prompts queries $\mathbf{Q}_T$) performs cross-attention with the extracted visual features. We use cross-attention to assign all possible relevant visual features to textual prompt queries, and use hard assignment for anatomical prompt queries. The reason for employing hard cross-attention is to ensure that each anatomical query gathers discriminative visual regions without overlaps. The effectiveness of the hard assignment is verified by the results in the following table.
> |Tumor Dice(%)|Liver|Pancreas|Hepatic Vessel|Colon|Colon in T4|
> |-----------------------------|:---:|:------:|:------------:|:---:|:---------:|
> |CAT w/othe hard assignment|72.18|46.46|69.97|46.65|58.49|
> |CAT w/othe PromptRefer mask |72.64|48.49|69.02|47.29|53.67|
> |CAT|72.73|49.67|70.11|48.31|57.37|
>
> In the PromptRefer, refined segmentation queries $\mathbf{Q'}_S$ engage in cross-attention with refined anatomical prompts queries $\mathbf{Q'}_A$ and textual prompts queries $\mathbf{Q'}_T$  to enhance segmentation. We employ a conventional attention mechanism, supplemented by carefully crafted attention masks, These masks force a group of prompt queries is employed to a specific segmentation query. This process aligns with empirical insights suggesting that accurately localizing typical tumors necessitates recognizing anomalous features within the pertinent organ. As can be seen from the table, this strategy helps to segment tumors that invade other organs (e.g., Stage-IV). We hope the above explanation and the provided code in Supplementary Material can address your confusion.
>
> # W3&W4: Clarification of Table 3 and Pre-trained models
> 1. Sorry for the confusion caused by the use of symbols (✓ and ✔️). As discussed in Section 4.2, the second-to-last row in Table 3, marked with a ✔️, indicates our use of hard assignment across all cross-attention layers. We conducted this experiment to validate our hypothesis that different types of prompts play distinct roles in aggregating visual features. We will clarify the symbol in the revised caption of Table 3.
> 2. The pre-trained model employed for anatomical prompts is the Swin UNETR. We utilize Clinical-Bert to encode textual prompts.
> # Questions:
> We hope the following responses will address your questions:
>
> **Q1&Q2**: We present examples of visual prompt in **Figure** (Figure 1 of the rebuttal PDF) and the details of textual prompt in **[Json](https://anonymous.4open.science/r/Reb/textual_prompts.json)**.
> We have added more comparisons in the case of training models on the dataset and report the number of parameters of CAT and those comparison models. The results are shown in the following tables.
>
> |MSD Tumor (Dice (%))|Liver|Pancreas|Hepatic Vessel|Colon|
> |--------------------| :----------: | :-------------: | :-------------------: | :----------: |
> |nnUNet |64.52 |43.80| 64.62 |40.44|
> |ZePT|66.53 |44.05| 66.18 |40.16|
> |CAT|**69.65** |**47.55**|**69.43**|**46.13** |
>
> We trained the CAT, ZePT, and nnUNet models on four MSD datasets. The results demonstrate that CAT still achieves superior outcomes even when trained on a single dataset, further validating our approach of integrating textual and visual prompts for segmentation.
>
> |Model|CAT|nnUNet|SAM-Med3D|SegVol|ZePT|
> |----------|:-------:|:-------:|:---------:|:-------:|:-------:|
> |Parameters|345.53M|235.02M|374.42M|673.86M|745.94M|
>
> **Q3**: In Figure 4 of our paper, we illustrate how visual and textual prompts enhance segmentation. Unfortunately, the heatmap format may have caused some misunderstanding. To address this, we provide a comparison in **Figure** (Figure 2 of the rebuttal PDF). While our textual prompts cover most scenarios, using them alone fails to encompass all regions. This observation further supports the claim that aligning textual and visual representations is challenging. Conversely, relying solely on anatomical prompts results in a high false positive rate and overly sharp boundaries. We hope these visual examples provide clearer clarification for you.
>
> # Limitations
> We appreciate your suggestions. To assess CAT’s efficacy in dealing with rare cases (i.e., the long-tailed problem), we introduce an in-house dataset where colon tumors have invaded adjacent organs. According to medical literature [3] [4],  T4 colorectal tumors, which represent only about 5-8.8% of colon tumor cases, pose significant challenges in diagnosis and treatment. Our results demonstrate that CAT significantly outperforms other models in segmenting T4 colon tumors, underscoring the effectiveness of our design in handling complex medical scenarios. We hope our early exploration could bring new insights into the field of the community.
>
> # References
> [1]  Clip-driven universal model for organ segmentation and tumor detection.
>
> [2]  Attention is all you need.
>
> [3]  Results after multi-visceral resections of locally advanced colorectal cancers: an analysis on clinical and pathological t4 tumors.
>
> [4]  Identification of risk factors for lymph node metastasis of colorectal cancer.

---

> > ### Comment · Reviewer_wEKt · 2024-08-12
> > **Nice rebuttal**
> >
> > Thanks for the excellent rebuttal. My confusion is solved.

---

> > > ### Author Response · Authors · 2024-08-12
> > >
> > > We sincerely appreciate your thorough review and constructive feedback on our manuscript. Your detailed comments have provided invaluable insights that have significantly contributed to the refinement of our work.

---

### Author Rebuttal · Authors · 2024-08-05

We sincerely appreciate all reviewers’ time and efforts in reviewing our paper. We are glad that reviewers are generally interested in our proposed method of combining anatomical and textual prompts for medical image segmentation. We also thank all reviewers for their insightful and constructive suggestions, which helped improve our paper further. To provide a clearer understanding, we update the backbone and motivation in our revised manuscript according to the comments. **All figures used in the rebuttal are shown in the attached PDF**.
# Backbone
**We hope that the illustration provided below will enhance the clarity and understanding of our paper.**

Previous promptable segmentation models in the medical domain can be categorized into textual-prompted methods and visual-prompted methods. Despite promising advancements, relying solely on visual or textual prompts has limitations. Textual-prompted methods utilize textual representations from referred text phrases to guide the segmentation process, requiring alignment between visual and textual representations. Although descriptive texts can cover intricate and rare anomalies with domain knowledge, data scarcity due to long-tailed distribution hinders the effective learning of alignments between textual and visual representations. This issue is particularly significant in the medical domain, where numerous corner cases (e.g., tumors with variations in shape, size, density distribution, and blurring boundaries) need to be addressed. Visual prompts, on the other hand, do not require cross-modal alignment and provide a more intuitive and direct method. However, they fail to convey the general concept of each object. For instance, tumors in different cancer stages exhibit diverse shapes and sizes, necessitating a comprehensive image collection to visually convey the abstract notion. In this work, we aim to develop a segmentation model that leverages the strengths of both visual and textual prompts without human interaction, striving for a fully automatic model for medical professionals.

Specifically, our visual prompts are derived from the relevant anatomical structures (i.e., cropped 3D CT images) and textual prompts are curated based on medical domain knowledge, as shown in the following links: **Anatomical prompts**(Figure 1 of the rebuttal PDF) and **[Textual Prompts](https://anonymous.4open.science/r/Reb/textual_prompts.json)**. For these two prompts, we apply different feature-gathering strategies. We use soft assignments to gather all possible relevant visual features for textual prompt queries and hard assignments to obtain discriminative visual regions without overlaps.
# Motivation of the Module Design
**We hope this section of our discussion clarifies the motivation behind our module design (the concerns raised by Reviewer qosx), specifically the rationale for our proposed ShareRefiner and PromptRefer.**

The underlying motivation of ShareRefiner module is to provide general concepts that encompass a wide range of scenarios within the medical domain via textual prompts and offer more intuitive and direct cues to mitigate the coarse visual-textual alignment issues via visual prompts. Consequently, we utilize soft assignment to assign all potentially relevant visual features to textual prompt queries while applying hard assignment for anatomical prompt queries to ensure that each anatomical query accurately captures discriminative visual regions without overlap. The rationale for designing attention masks in PromptRefer is to direct queries to focus specifically on relevant objects and prevent the introduction of noise from irrelevant regions. In clinical practice, accurately localizing the typical tumor requires being aware of the anomalous features in the relevant organ, and even identifying organs requires focusing on the anatomical structures involved.  This strategy aligns with practical experience, which suggests that effective segmentation of target objects necessitates a heightened focus on the relevant contextual details.

# Contributions
1. We present a promising attempt toward comprehensive medical segmentation via coordinating anatomical-textual prompts. Apart from performing generic organ segmentation, our model can identify varying tumors without human interaction.

2. To effectively integrate two prompt modalities into a single model, we design ShareRefiner to refine latent prompt queries with different strategies and introduce PromptRefer with specific attention masks to assign prompts to segmentation queries for mask prediction.

3. Extensive experiments indicate that the coordination of these two prompt modalities yields competitive performance on organ and tumor segmentation benchmarks. Further studies revealed robust generalization capabilities to segment tumors in different cancer stages.

4. We highlight several critical challenges in medical image segmentation and call for further research on utilizing both textual and visual prompts to address intricate scenarios. We hope our model can support professionals in the arduous clinical diagnosis process.

---

### Decision · Program_Chairs · 2024-09-25

**Decision:**

Accept (poster)

**Comment:**

The paper obtains positive reviews from three knowledgeable reviewers. After rebuttal and discussions, the three reviewers rate the paper as 1 'Weak Accept' and 2 'Borderline Accept‘. Overall, the reviewers are happy about the key idea of coordinating anatomica and textual prompts for better organ and tumor segmentation as well as the experimental evaluation.

I recommend the acceptance of the paper.